

# Design and performance of the Cluster Ion Counter (CIC)

Sander Mirme[1,2], Rima Balbaaki[3], Hanna Elina Manninen[4,3], Paap Koemets[1,2], Eva Sommer[4,5], Birte Rörup[3], Yusheng Wu[3], Joao Almeida[4,6], Sebastian Ehrhart[7], Stefan Karl Weber[4], Joschka Pfeifer[4], Juha Kangasluoma[3], Markku Kulmala[3], and Jasper Kirkby[4]

[1]Institute of Physics, University of Tartu, W. Ostwaldi 1, 50411 Tartu, Estonia
[2]Airel OÜ, Observatooriumi 5, 61602 Tõravere, Estonia
[3]Institute for Atmospheric and Earth System Research (INAR), University of Helsinki, 00014 Helsinki, Finland
[4]CERN, European Organization for Nuclear Research, CH-1211 Geneve 23, Switzerland
[5]Faculty of Physics, University of Vienna, 1090 Vienna, Austria
[6]Faculty of Sciences of the University of Lisbon, 1749-016 Lisbon, Portugal
[7]Max Planck Institute for Chemistry, Hahn-Meitner-Weg 1, 55128 Mainz, Germany

**Correspondence:** Sander Mirme (sander.mirme@ut.ee)

**Abstract.** A dilute plasma is continuously maintained in the troposphere by ionising particle radiation from galactic cosmic rays and radon decay. Small ions in the 1–2 nm size range play an important role in atmospheric processes such as ion-induced nucleation of aerosol particles. Consequently there is a need for precise and robust instruments to measure small ions both for atmospheric observations and for laboratory experiments that simulate the atmosphere. Here we describe the design and performance of the Cluster Ion Counter (CIC, Airel OÜ), which simultaneously measures the number concentrations of positively- and negatively-charged ions and particles below 5 nm mobility diameter, with low noise and fast time response. The detection efficiency is above 80% for ions and charged particles between 1.2 and 2.0 nm, and above 90% between 2.0 and 3.0 nm. The ion concentrations measured by the CIC agree well with reference instruments. The noise level ($1\sigma$ of background measurements) is typically between 20 and 30 ions cm$^{-3}$ at 1 Hz sampling rate and an air flow rate of 7 l min$^{-1}$ per analyzer. The noise level improves when higher flow rates and longer sampling periods are used. The CIC responds rapidly with 1 s time resolution to pulses of ionisation produced in the CLOUD chamber by a CERN particle beam.

## 1 Introduction

Atmospheric ions are ubiquitous throughout the atmosphere, with ion pair concentrations in the troposphere ranging between a few 100 cm$^{-3}$ at ground level (Hirsikko et al., 2011) and up to around 2,000 cm$^{-3}$ at the tropopause. In the troposphere, primary ions of nitrogen and oxygen are generated by galactic cosmic rays and ionising radiation from the decay of radon of other radioactive isotopes. The charge is rapidly transferred by molecular collisions to trace vapours of higher gas-phase acidity or proton affinity to form so-called small ions, which comprise both charged molecules and charged molecular clusters. The ion-ion recombination lifetime of small ions in the troposphere is around ten minutes, but the actual lifetime may be shorter due to additional loss mechanisms such as scavenging by pre-existing particles, i.e. diffusion-charging of aerosol particles. An early classification of air ions (Hõrrak et al., 2000) defined small ions (also called cluster ions) as those with mobility diameter





below 1.6 nm, intermediate ions between 1.6 and 7.5 nm, and large ions between 7.5 and 20 nm. However, contemporary studies would classify air ions above the size of small ions as charged aerosol particles since they contain more than around ten molecules and are effectively stable particles.

Despite their short lifetime, small ions play a key role in atmospheric electricity, clouds and climate. The drift of small ions in the electric field between the ionosphere and Earth's surface gives rise to a continuous fair-weather current density of $2 \, \mathrm{pA \, m^{-2}}$ flowing through the global electrical circuit (Rycroft et al., 2012). Under certain conditions, ions can enhance 10–100-fold the nucleation rate of aerosol particles, which may then grow by condensation of ambient low volatility vapours to sizes where they become cloud condensation nuclei (Kirkby et al., 2011, 2016, 2023). Ions and charged aerosol particles may also directly affect cloud microphysical processes and, in turn, climate (Carslaw et al., 2002; Harrison and Carslaw, 2003; Tinsley, 2008; Harrison and Ambaum, 2008).

Early indications that free ions existed in air were inferred from the decay of charge from gold leaf electroscopes. By this means—and only three years after J.J. Thomson had discovered the electron—C.T.R. Wilson calculated that 20 ion pairs $\mathrm{cm^{-3} s^{-1}}$ were being continuosly produced in air (Wilson, 1900). This remarkable precision is within a factor two of the ionisation rate from galactic cosmic rays, which were unknown at the time. The original electronic instrument to measure air ion concentrations (air conductivity) was an aspiration counter developed in the early 1900's by Ebert (1901) and Gerdien (1905). The same era witnessed various designs of aspiration counters, first order differential counters, second order differential counters and counters with several collecting electrodes. A detailed description of these counters and the history of their development can be found in Tammet (1970) and Flagan (1998).

The principle of operation of the aspiration counter is illustrated by the aspirated coaxial cylinder condenser (Flagan, 1998). As the name suggests, this instrument consists of two concentric cylindrical electrodes, a device to draw in air (an aspirator, a fan or a pump), and an electrometer. One of the electrodes is connected to a voltage source to create an electric field between the inner and outer electrodes. As air is drawn through the device, the ions drift towards the collecting electrode of opposite polarity and strike it. The ion number concentration is determined from the flow rate and the current passing from the collecting electrode to ground, which is measured with a sensitive electrometer. The geometric design of the device, the operational voltage and the flow rate determines the range of electrical mobilities of the air ions that are collected, and thus their size range.

Subsequent designs of aspiration counters have developed automatic data recorders, programmable control system, improved flow control, better electrometer signal-to-noise and higher time resolution (Moody, 1984; Aplin and Harrison, 2000; Grabarczyk, 2001; Harrison and Wilding, 2005; Nicoll and Harrison, 2008; Kubásek et al., 2009). Various names are used in the literature for the aspirated coaxial cylinder condenser, such as the Ebert ion counter, the Gerdien aspirator, the aspirated Gerdien condenser, the aspiration condenser, the ventilated cylindrical capacitor, the cluster ion counter, and the air ion counter.

While recent air ion counters have progressed considerably beyond early designs, several limitations persist. One notable limitation of many devices is their inability to simultaneously detect positive and negative ions. Furthermore, there is a general lack of proper laboratory verification of the instrument's detection efficiency and the mobility ranges of the collected ions. Another challenge is the high fixed flow rate of most devices, which is a problem for laboratory experiments with restrictions on maximum flow rates. Furthermore, the simple cylindrical design of many current devices makes integration into experimental



setups difficult and limits their compatibility with space constraints of typical measurement containers. Moreover, long-term air ion monitoring and atmospheric research requires high-precision instruments with low noise and fast time response, and this performance is not provided by the basic handheld devices that predominate in the commercial market.

Here we describe the design and performance of a novel air ion counter—the Cluster Ion Counter (CIC, Airel OÜ)—that
addresses the limitations outlined above. Its design builds on the earlier ion counters developed at Tartu University since the 1950s (Matisen et al., 1992). The key design requirements are improved time resolution, reduced ion losses and higher sensitivity at low flow rates. We will first present the theoretical principles and design of the CIC. We will then describe calibration of the CIC in the laboratory and its performance at the CERN CLOUD chamber, where it has been in operation since 2018.

## 2   Methods

### 2.1   Instrument design

The CIC measures the number concentrations of positive and negative air ions simultaneously. It comprises two independent cylindrical differential mobility analyzers, each with three collecting electrodes (Fig. 1 and Fig. 2). The ions in the air entering the analyzers are repelled by a central electrode, which is held at a fixed potential between $-40$ and $40\,\mathrm{V}$. A switchable high
voltage ($> 120\,\mathrm{V}$) electrical inlet filter is used to prevent ions from entering the analyzer during zero-point measurements. Charged particles deposit on the outer wall of the analyzer which is divided into three independent collecting electrodes. The current flowing through each electrode is measured with a high-precision integrating electrometer that samples and digitizes the current at up to 30 Hz. The inlet of the analyzer has been designed to minimize ion losses from diffusion and stray electric fields. A flow meter and blower at the exit of each analyzer controls the flow rate of the air drawn through the instrument.

The geometry of the analyzer has been chosen so that, for $10\,\mathrm{l\,min^{-1}}$ flow rate and $6.5\,\mathrm{V}$ voltage on the central electrode, the limiting mobilities of the three collecting sections are 2.5, 0.5 and $0.25\,\mathrm{cm^2V^{-1}s^{-1}}$. As it can be seen from Fig. 3, ideally, all ions with a mobility $z > 2.5\,\mathrm{cm^2V^{-1}s^{-1}}$ ($d_p < 0.9\,\mathrm{nm}$) are deposited on the first electrode, $z > 0.5\,\mathrm{cm^2V^{-1}s^{-1}}$ ($d_p < 2.0\,\mathrm{nm}$) on the first and second, and $z > 0.25\,\mathrm{cm^2V^{-1}s^{-1}}$ ($d_p < 2.8\,\mathrm{nm}$) on the first, second and third electrode. The limiting mobilities are primarily determined by the geometry of the analyzer and the ratio between air flow rate and central electrode voltage. The air
flow rate through the CIC can be selected by software to values between 5 and $60\,\mathrm{l\,min^{-1}}$ per analyzer. The central electrode voltages are automatically adjusted so that the ratio (and consequently the limiting mobilities of the collecting electrodes) remains constant.

Figure 4 shows the theoretical transfer functions for the CIC analyzer based on a simple general model of a first order differential mobility analyzer. This does not take into account diffusional and electrical losses, diffusional dispersion, as well as
specific airflow characteristics caused by mechanical imperfections. The data analysis uses experimentally-determined transfer functions, which are broader and flatter than those predicted by the simple theoretical model. The electrodes do not have a sharp lower cut-off limiting mobility. Due to the absence of a sheath air flow, electrodes will always detect a small fraction of larger particles with low mobility that enter the analyzer close to the collecting electrode. It is possible to estimate the cluster



ion concentrations with a sharp cut-off mobility between 0.5 and $0.25\,\mathrm{cm^2V^{-1}s^{-1}}$ by estimating and subtracting the signal
contributed by larger particles based on the signal from the third electrode.

The measurement of small currents (typically 1–10 fA) generated by air ions is demanding and causes many problems related to change of temperature, relative humidity, turbulence, and electromagnetic noise in ambient conditions. The design of the electrometers, and data acquisition electronics as well as signal processing algorithms of the CIC are identical to that of the Neutral cluster and Air Ion Spectrometer (NAIS; Mirme, 2011; Mirme and Mirme, 2013). Each collecting electrode of the CIC
is connected to its own dedicated, low-noise, integrating electrometer. The electric current produced by the depositing ions is accumulated on a capacitor (typically 33 pF) in an integrator circuit using a low input current operational amplifier (LMC6042). The the output voltage of the electrometer, which is proportional to the accumulated charge, is periodically measured by a 24-bit analog-to-digital converter. The average electric current is calculated from the rate of change of the voltage. Since ion current collection is continuous and independent of the analog-to-digital converter's sampling frequency, the integration time can be
optimized based on the desired signal-to-noise ratio and time resolution. To prevent saturation, the capacitor is automatically discharged and reset to zero volts when the electrometer output voltage reaches a threshold, typically ±1 V. This reset causes a brief 5 to 10 second gap in the measurements, occurring usually once every hour to once every day.

Furthermore, the CIC employs a switchable high voltage electrical inlet filter to block ions from entering the analyzer during calibrations of zero-level currents and noise levels of the electrometers. The device automatically carries out this periodic
calibration to correct the measured signals for zero drift, to monitor the health of the device and to indicate the possible need for maintenance. Zero-level, or so called "offset" measurements, are typically run for 20–30 s during every 1–5 minutes of signal measurements. The low ion losses and high measurement rate of the electrometers allow the instrument to achieve very fast time resolution, above 1 Hz.

The CIC uses internal Venturi flow meters and blowers independently for both analyzers in a software-controlled feedback
loop to maintain constant sample airflow rates. Temperature, relative humidity and absolute air pressure are also monitored for both air flows. The central electrode voltages are supplied by a bipolar software-controlled voltage source, which allow each analyzer to operate at a freely-chosen voltage and polarity. To confirm correct operation of the device, all relevant parameters from the different subsystems are monitored and logged alongside the measurement data. A graphical measurement software (Spectops) is provided for monitoring, controlling and storing measurement and diagnostic data from the CIC. A quick-view
program (Retrospect) can be used to review the stored data files.

## 2.2   Parameters measured by the CIC

### 2.2.1   Number concentration of small ions

Ion number concentrations are calculated from the current on each of the three collecting electrodes, which is proportional to the ion deposition rate. The current I (A) measured by each electrode is related to the sampled ion concentration $N\pm$ (ions · cm$^{-3}$)
as follows,

$$I = N_{\pm} n e Q_s \tag{1}$$





where $n$ is the average number of elementary charge units per ion (assumed to be unity), $e = 1.6 \times 10^{-19}$ C is the elementary unit of charge, and $Q_s$ (cm$^3$s$^{-1}$) is the volumetric sample flow rate passing the electrode. The summed signal from the three electrodes in each analyzer, together with Eq. 1, provides the total concentration of positive and negative ions below 5 nm diameter.

### 2.2.2 Mobility diameter of small ions

Electrical mobilities of ions and charged particles can be converted to particle diameters through the Millikan-Fuchs relation,

$$Z_p = \frac{neC_c}{3\pi\eta d_p} \tag{2}$$

as described in Mäkelä et al. (1996) and also Ku and de la Mora (2009), where $Z_p$ is the electrical mobility, $n$ the number of elementary charges, $e$ the elementary charge, $C_c$ the Cunningham slip correction factor, $\eta$ the dynamic viscosity of gas, and $d_p$ the particle diameters. At these small sizes, all particles can be assumed to be singly charged.

### 2.2.3 Air conductivity

The conductivity of air is mostly due to ions with the highest electrical mobility, i.e. small ions. However, larger charged particles also contribute. According to a long-term study in Hõrrak (2001), small ions were responsible for 96.3% of the total conductivity of air. Thus, the small ion concentrations $N_+$ and $N_-$ measured by the CIC can provide a measurement of the polar conductivities of air, $\lambda_+$ and $\lambda_-$, by the equations,

$$N_+ \approx \frac{0.96\lambda_+}{Z_+e} \quad \text{and} \quad N_- \approx \frac{0.96\lambda_-}{Z_-e} \tag{3}$$

where $Z_\pm$ is the mean electric mobility measured by the CIC and $e$ is the elementary charge (Hirsikko et al., 2011).

### 2.3 Experimental methods used to evaluate the performance of the CIC

Experimental evaluations of the CIC have been carried out in several aerosol laboratories: at the Institute for Atmospheric and Earth System Research (INAR), University of Helsinki, at the CERN CLOUD chamber, Switzerland, at the Laboratory of Atmospheric and Environmental Sciences, University of Tartu, and at Airel OÜ, Tartu.

At INAR, Universty of Tartu and Airel, the calibration aerosol were generated and size-selected in fixed, ultra-dry conditions. At CERN, the calibration aerosol were nucleated and grown from various atmospherically-relevant vapours in the CLOUD chamber (Kirkby et al., 2023), the contents of which were continuously sampled by a suite of instruments that included the Airel CIC and NAIS (Fig. 5). The calibrations at INAR supported ion cluster experiments performed at CLOUD. Thus, in both locations, the sampling line was the same as used at CLOUD, which included a Y-splitter for connection to the CLOUD sampling probe. On the other hand, at the University of Tartu and at Airel, the CIC was calibrated without a CLOUD-specific inlet, thereby providing measurements that are more applicable to general operation.





### 2.3.1 Calibration of ion number concentrations and mobility diameters

At the University of Tartu and at Airel, ions were generated using a tungsten glowing-wire source operating at nitrogen flow rates from 1.5 to 4.0 $l\,min^{-1}$. The sample was neutralized using a soft X-ray source and particles were size selected using a Vienna-type DMA (Uin et al., 2006) built at the University of Tartu.

Calibration experiments at INAR focused on concentration and mobility measurements of small ions between 0.8 and 5.0 nm. Small ions were generated with a tungsten-wire aerosol generator operated at 15 $l\,min^{-1}$ nitrogen flow (Kangasluoma et al., 2015). The sample mobility was selected with a Herrmann-type high resolution DMA (HDMA). Prior to the mobility selection, the sample was neutralized with an $^{241}$Am radioactive source. We carried out measurements with both positive and negative ions, to confirm equal response of the two mobility analyzers in the CIC.

For measurements carried out at CERN and at INAR, the CIC was sampling with a constant flow rate of 7.0 $l\,min^{-1}$ per analyzer. The reference for the number concentration was provided by a TSI Electrometer 3068B, at 2.5 $l\,min^{-1}$ sample flow rate and ±1 fA current accuracy (±150 $ions\,cm^{-3}$). The measured concentrations were corrected for diffusional losses in the sampling lines (Gormley and Kennedy, 1948). The calibration set-up is described in detail in Wagner et al. (2016).

At Tartu, most of the experiments were done at 30 $l\,min^{-1}$ flow rate per analyzer through the CIC. Additional experiments were carried out to investigate the effect on ion detection efficiency of sample flow rates in the range from 10 to 50 $l\,min^{-1}$.

### 2.3.2 Evaluation of instrument background and time response

Experiments at the CERN CLOUD chamber provided measurements of the CIC background and time response. The CIC background, i.e. finite electrometer signals when no ions are present in the sampled air, was measured during periods when a high voltage was applied to electrodes inside the CLOUD chamber, which sweeps out small ions in less than 1 s. The time response of the CIC was evaluated using the pion beam from the CERN Proton Synchrotron, which arrives in short pulses of 0.4 s duration separated by around 30 s.

## 3 Performance of the CIC

### 3.1 Detection efficiency in sub-5 nm diameter range

We measured the detection efficiencies of the CIC for 1.2–5.0 nm positive and negative ions for each of the 3 collecting electrodes, as well as a sum of all 3 electrodes (Fig. 6). The total ion detection efficiency exceeds 80% between 1.2 and 2.0 nm mobility diameter and 90% between 2.0 and 3.0 nm mobility diameter. The detection efficiency decreases above 3.0 nm, which provides a natural cutoff for the CIC outside the range of small ions.

We also investigated the influence on ion detection efficiency of various CIC sample flow rates at 10, 30 and 50 $l\,min^{-1}$ (Fig. 7). The higher sample flow rates produce mild increases in the detection efficiency below 3.0 nm, and slight decreases for larger particles. Since higher flow rates provide an improved signal-to-noise ratio, it is generally preferable to use the maximum





sample flow rate within the constraints of the experimental setup. If the sampling inlet includes bends or a Y splitter, then the choice of optimum flow rate also needs to take into account turbulence at high flow rates, which leads to additional losses.

## 3.2 Linearity of the CIC response

Figure 8 shows calibration measurements made with charged tungsten particles and the HDMA setup in the sub-5 nm size range. We compared the ion concentrations measured by the CIC with those of a reference electrometer. The linearity was

measured for 0.9, 1.5 and 2.3 nm positive ions and for 0.9, 2.1 and 2.9 nm negative ions. In all cases, the response was linear for ion concentrations between $10^2$ and $10^5$ cm$^{-3}$. These laboratory calibrations included the same Y splitter on the inlet as used at the CLOUD chamber, which introduces additional losses. Therefore the CIC efficiency cannot be directly inferred from these data. As well as demonstrating excellent linearity, these measurements show that the CIC can be operated reliably in environments from pristine to highly-polluted i.e. aerosol number concentrations of up to $10^5$ cm$^{-3}$.

## 3.3 CIC noise level

When the CIC measures in ion-free conditions, a small background ion concentration is always recorded due to unavoidable electronic noise in the electrometers. This sets a threshold ion concentration below which the CIC is unable to provide useful measurements. Fig. 9 and Table 1 summarise CIC measurements at CLOUD when the electric clearing field is present, i.e. all ions are swept from the chamber in less than 1 s. The CIC was sampling the chamber air with a flow rate of 7.0 l min$^{-1}$ per

analyzer, with an uncertainty of ±5%. Under these conditions, and for 1 s integration time, the median value of the instrument background for the ion concentration is below ±5 cm$^{-3}$ (±0.1 fA). The 25th and 75th percentiles are below ±20 cm$^{-3}$ (±0.4 fA). The 5th and 95th percentiles are between ±20 and ±50 cm$^{-3}$ (±0.4 ··· ±0.9 fA). Standard deviation ranges from 18 to 35 cm$^{-3}$ (0.35 to 0.65 fA) with a median at 26 cm$^{-3}$ (0.48 fA). Lower noise levels can be reached by averaging signals over times longer than 1 s or by using higher sample flow rates. This instrument background is excellent and substantially smaller than most

previous air ion spectrometers or electrometers that are commonly used as reference instruments.

## 3.4 CIC time response

The time response of the CIC has been measured at the CLOUD chamber using the CERN particle beam (Fig. 10). The pion beam spill from the CERN Proton Synchrotron arrives in short bursts of 0.4 s duration, which, for these data, are spaced by about 30 s and have a flux of $1.3 \times 10^6$ particles/burst (Fig. 10b). The CIC ion number concentration responds to each beam

pulse after a 5 s delay (Fig. 10a). This delay is due to the circulation time in the CLOUD chamber from the internal mixing fans for the beam-exposed air parcel to reach the tip of the CIC sampling probe. The CIC then reacts instantly (<1 s) to the peak ionisation pulse from the beam, followed by a gradual decay as the ions are mixed and diluted with non-irradiated air in the chamber. On longer time scales, ions are lost by ion-ion recombination and by collision with the walls of the CLOUD chamber. The response of the CIC to the beam bursts is mainly seen in channel 1 since only small ions are created in the chamber on

these short time scales (note the absence of any signal in channel 3, which has zero response for small ions below 2 nm). The



gaps in the CIC data correspond to periods when the instrument performed automatic internal zero-level measurements, with the inlet filter switched to 120 V. These data demonstrate an excellent response time for the CIC of around 1 s.

## 4 Summary

The laboratory measurements described here—along with operational experience at the CERN CLOUD chamber over the last five years—demonstrate that the CIC provides precise and robust long-term measurements of small ion concentrations of both polarities, with low noise, fast time response and excellent reliability. The instrument maintains precise small ion measurements in all environments from pristine to highly polluted. The low noise of the instrument—even at low flow rates—combined with a fast time response of 1 s, opens up new applications for small-ion counters. For example, in chamber studies, the CIC can precisely measure fast ion processes such as ionisation pulses, ion-ion recombination rates, diffusion charging of aerosol, ion-droplet interactions and wall loss rates. In the atmosphere, ion concentrations can be measured in rapidly-changing environments such as inhomogeneous urban conditions or observations from aircraft. In addition, the CIC can lead to an improved understanding of the production, transport and loss of small ions in the troposphere and their contributions to atmospheric processes such as the global electrical circuit and ion-induced nucleation of aerosol particles. With its excellent performance for small-ion measurements, the CIC will join other key instruments for studying atmospheric new particle formation such as the Neutral cluster and Air Ion Spectrometer (NAIS), the scanning Particle Size Magnifier (PSM) and the Atmospheric Pressure interface Time Of Flight (APi-TOF) mass spectrometer.

*Author contributions.* SM and PK designed the instrument. SM, HEM, PK, RB, ES, BR, YW, SE, SKW, SP, JP, JK, and JA performed measurements. SM, HEM, and RB analyzed the data. SM, HEM, RB wrote the manuscript draft. RB, BR, and JK reviewed and edited the manuscript. MK and JK facilitated access to research infrastructure.

*Competing interests.* Sander Mirme and Paap Koemets work for Airel OÜ. Sander Mirme is a shareholder of Airel OÜ.

*Acknowledgements.* This research has been in part supported by Estonian Research Council (projects PRG714, RVTT3), and MSCA ITN project CLOUD-DOC (grant number 101073026). RB acknowledges funding from the EMME-CARE project, which was supported by the European Union's Horizon 2020 Research and Innovation Programme (grant agreement no. 856612) and the Cyprus Government.



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





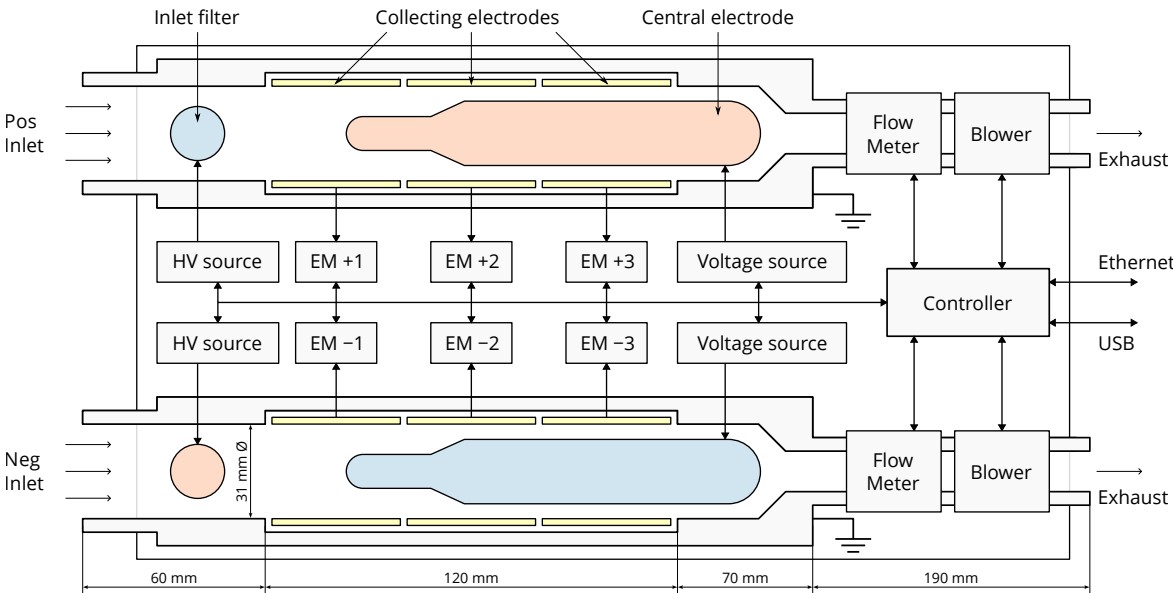

**Figure 1. Cross section and schematic of the Cluster Ion Counter (CIC).** The CIC has two identical analyzers to measure simultaneously positive ions and negative ions. Each analyzer has three collecting electrodes which measure different ranges of mobility diameters. Red shading indicates positive polarity and blue indicates negative polarity.

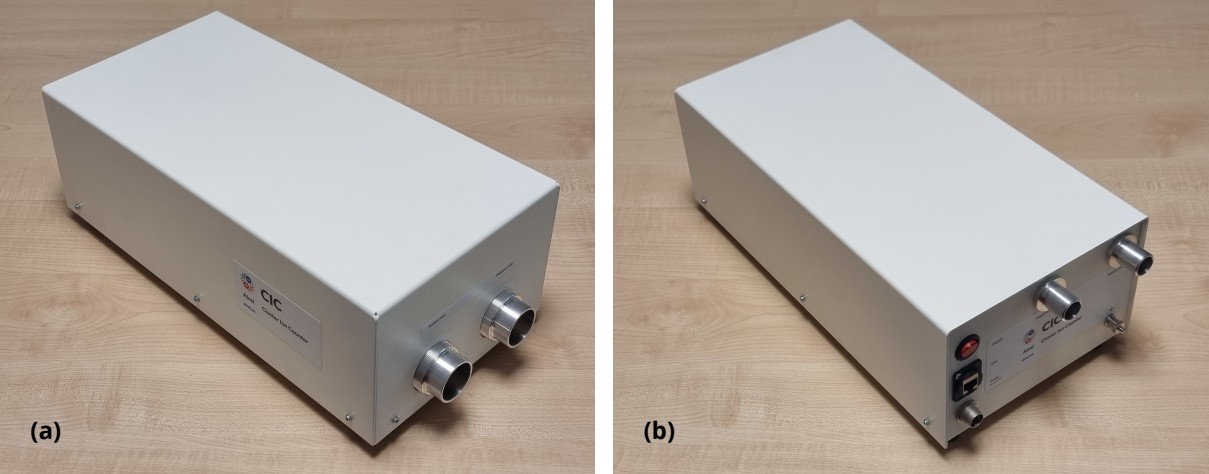

**Figure 2. Photos of the CIC.** The front view (a) of the CIC showing the two inlets with 35 mm outer diameter, and the back view (b) showing the outlets, power and communication connections. The dimensions of the CIC are 200 mm width, 130 mm height and 400 mm length.





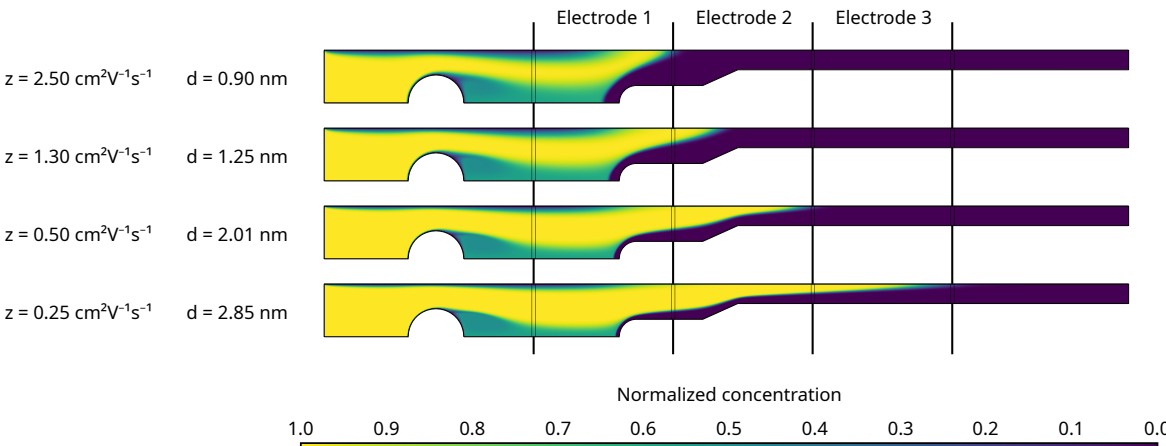

**Figure 3. Simulated flow patterns for singly-charged ions and particles.** Normalized concentration of singly-charged positive ions and particles of various diameters inside the CIC, according to a numerical model simulations. The air flow rate is $10\,\mathrm{l\,min^{-1}}$, the central electrode is at $6.5\,\mathrm{V}$ and the inlet filter is at $0\,\mathrm{V}$.





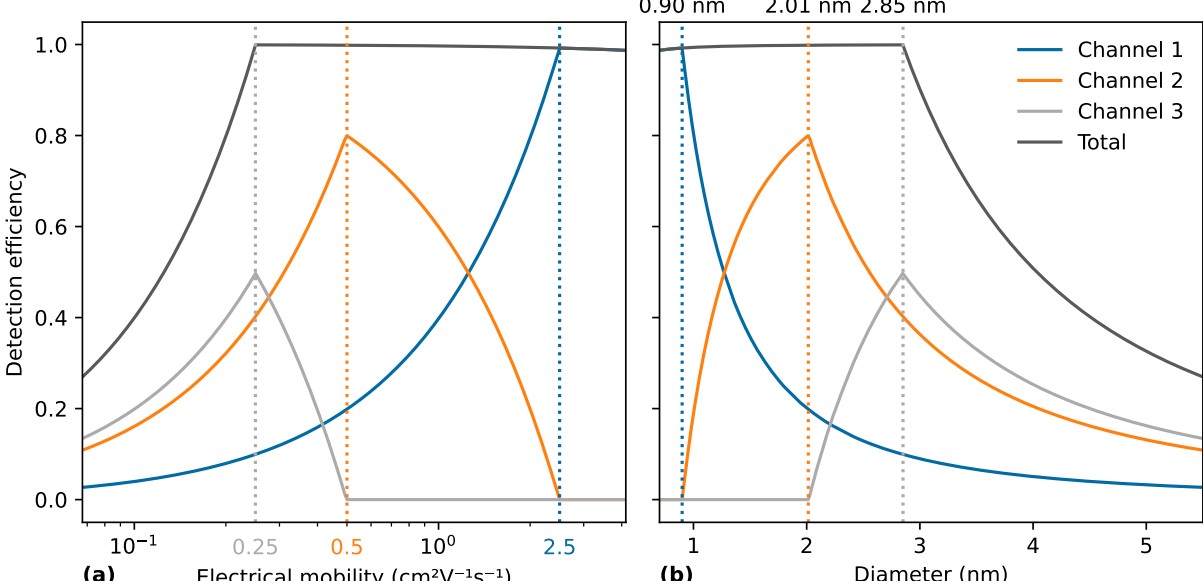

**Figure 4. Theoretical detection efficiencies of the three collecting electrodes versus diameter and electrical mobility.** Theoretical transfer functions of the three collecting electrodes, without considering ion diffusion losses, versus (a) electrical mobility and (b) mobility diameter. The detection efficiency indicates the ion flux reaching the collecting electrodes relative to the flux that enters the the CIC. The electrical mobilities of peak collection efficiency in the three sections are 2.5, 0.5 and $0.25\,\mathrm{cm^2V^{-1}s^{-1}}$ which correspond to mobility equivalent diameters 0.90, 2.01 and 2.85 nm. The summed response of the three collecting electrodes (red line) is calculated to provide a high and uniform detection efficiency for mobility diameters between 0.8 and 2.8 nm. The ratio of the central electrode voltage to air flow rate in a single analyzer is $6.5\,\mathrm{V}/10\,\mathrm{l\,min^{-1}} = 0.65\,\mathrm{V\,min\,l^{-1}}$





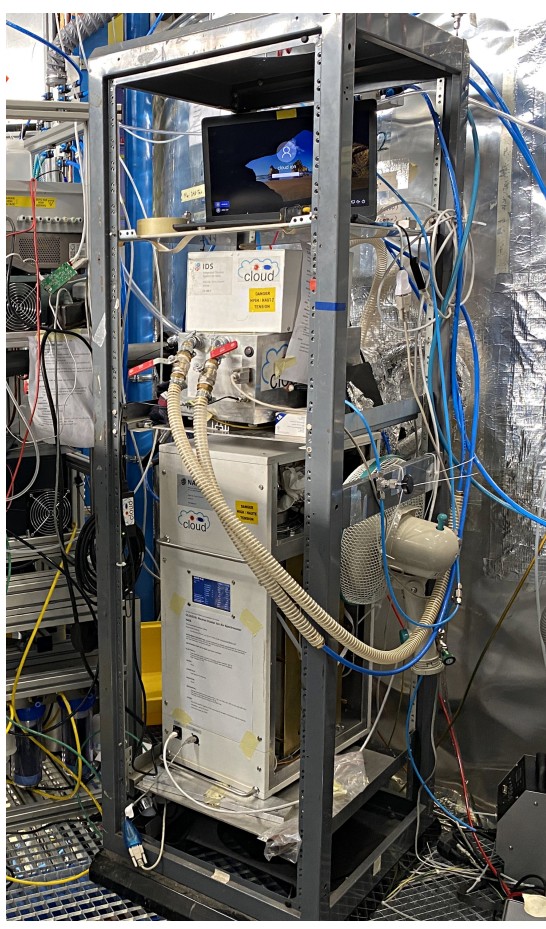

**Figure 5. The Airel CIC and NAIS at the CERN CLOUD chamber, November 2023.** The CIC is housed in the aluminium box with two prominent exhaust pipes with red hand valves at the exit of the instrument. The Airel Neutral Cluster and Air Ion Spectrometer (NAIS) is housed in the larger white cabinet below. The aluminised wall behind the instruments is the outer surface of the thermal housing for the CLOUD chamber. The two instruments are connected to the same CLOUD sampling probe via a Y-splitter inlet (not visible in the image).



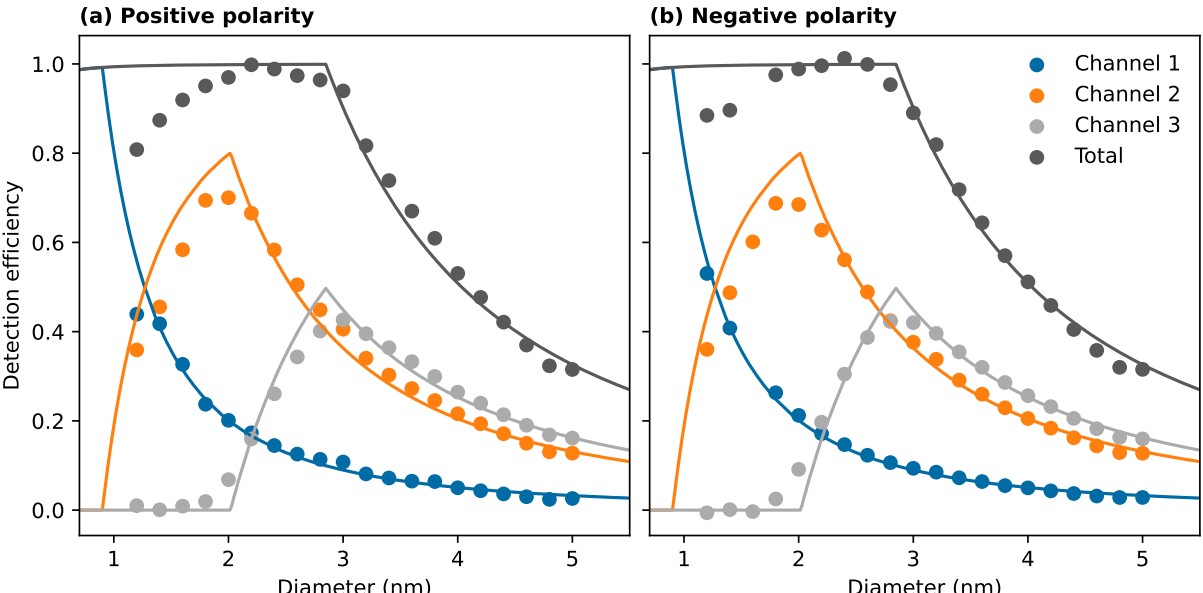

**Figure 6. Experimental detection efficiencies for the three collecting electrodes versus particle diameter and polarity.** Experimental detection efficiencies for 1.2–5.0 nm (a) positive and (b) negative ions and charged particles for each of the three collecting electrodes and for the sum of all three electrodes. The lines correspond to the theoretical model predictions. The discrepancy at small diameters between the measurements and the model is due to diffusion losses inside the instrument, which are ignored in the simulation.



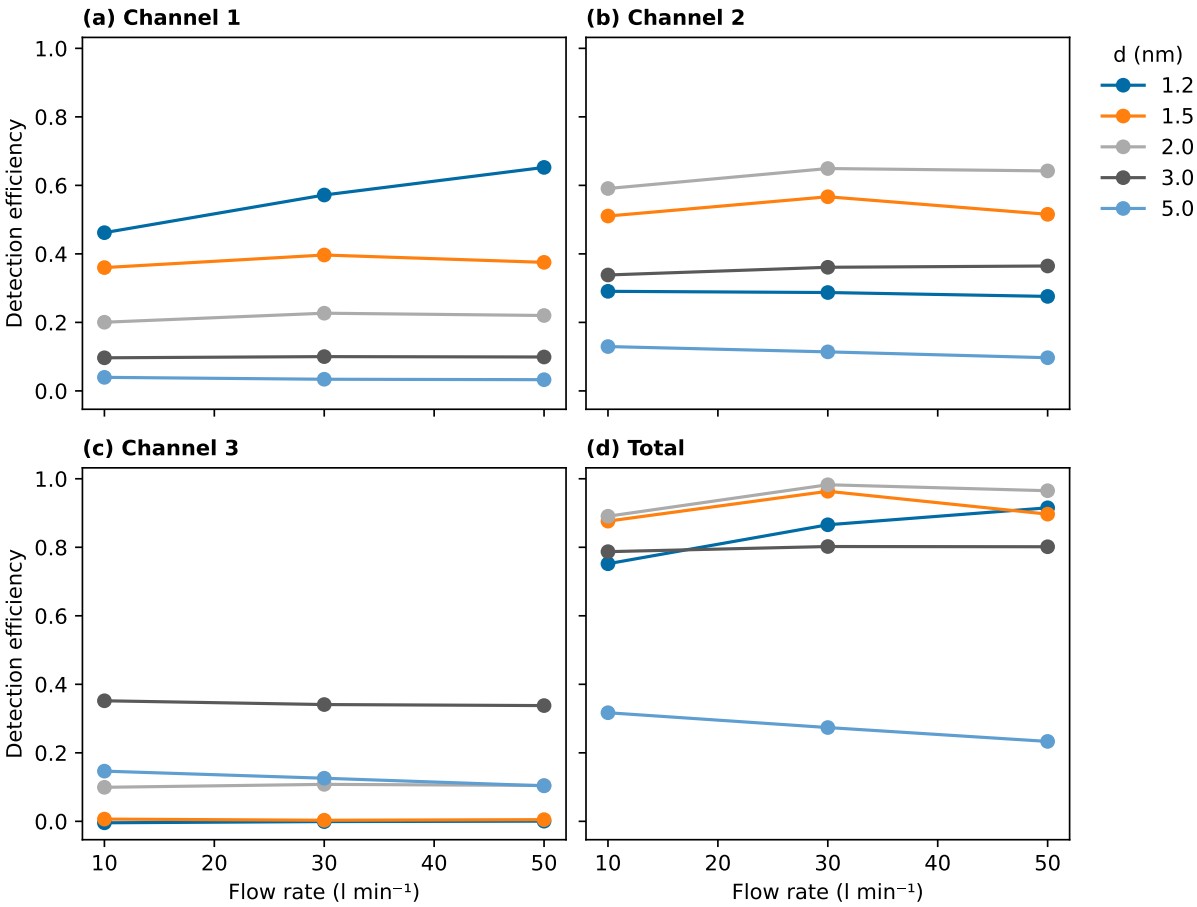

**Figure 7. Experimental detection efficiencies versus flow rate.** Counting efficiency versus CIC sample flow rate of one analyzer (half of total sample flow rate) for charged particles of various mobility diameters measured in (a) channel 1, (b) channel 2, (c) channel 3 and (d) summed channels 1–3. The ratio of the central electrode voltage to air flow rate in a single analyzer is kept at $0.65\,\mathrm{V\,min\,l^{-1}}$ to maintain a constant transfer function for each channel. The CIC can operate with high detection efficiency down to $10\,\mathrm{l\,min^{-1}}$ flow rate.

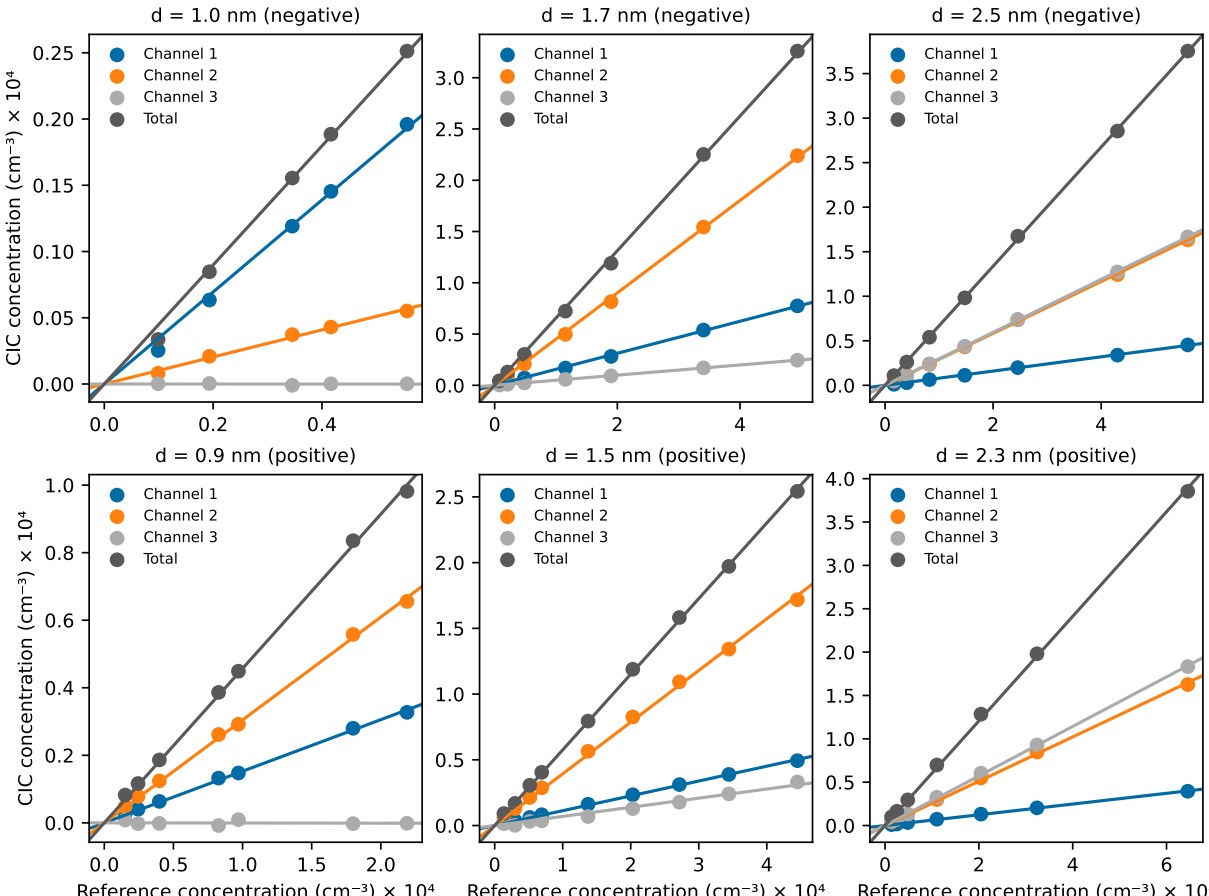

**Figure 8. Calibration of the CIC linearity.** Concentrations of ions and charged particles measured by the CIC versus reference electrometer concentrations for negative particles of diameter (a) 1.0 nm, (b) 1.7 nm and (c) 2.5 nm, and positive particles of diameter (d) 0.9 nm, (e) 1.5 nm and (f) 2.3 nm. The four lines in each panel correspond to channel 1 (blue), channel 2 (orange), channel 3 (green) and summed channels 1–3 (red). The data show excellent linearity and low noise for the CIC. The efficiency of the CIC cannot be inferred from these measurements since they were performed with an additional inlet that reduced the overall detection efficiency.

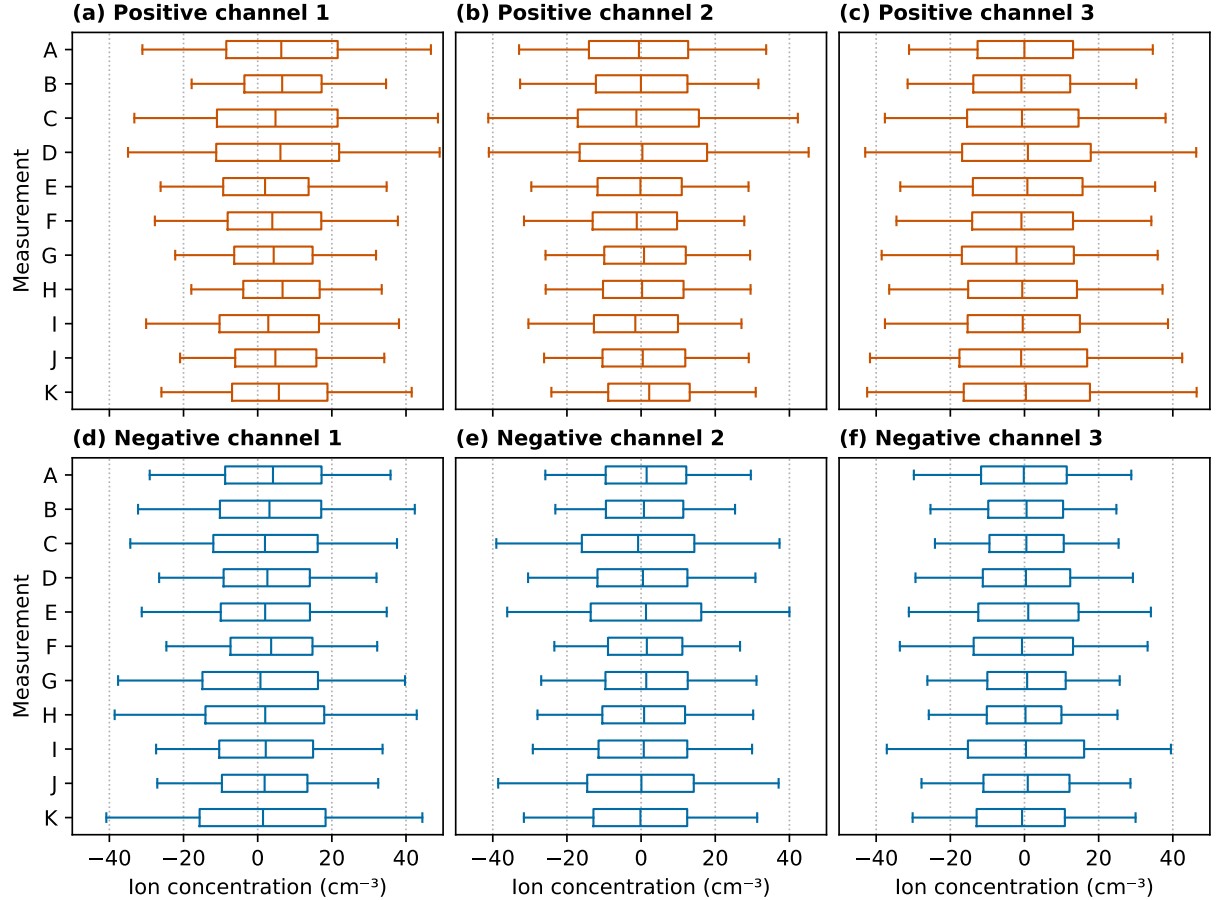

**Figure 9. Background measurements.** CIC background measurements with varying trace gas, temperature and relative humidity conditions in the CLOUD chamber when the high voltage clearing field was active and no charged particles were present inside the chamber. The box and whisker plots indicate indicate 25th–75th percentile range (boxes), 5th–95th percentile range (whiskers) and median values (lines in the middle of boxes) of CIC 1 second average samples. The CIC was measuring at $7\,\mathrm{l\,min^{-1}}$ sample flow rate per analyzer. The corresponding chamber conditions for each measurement are shown in Table 1.

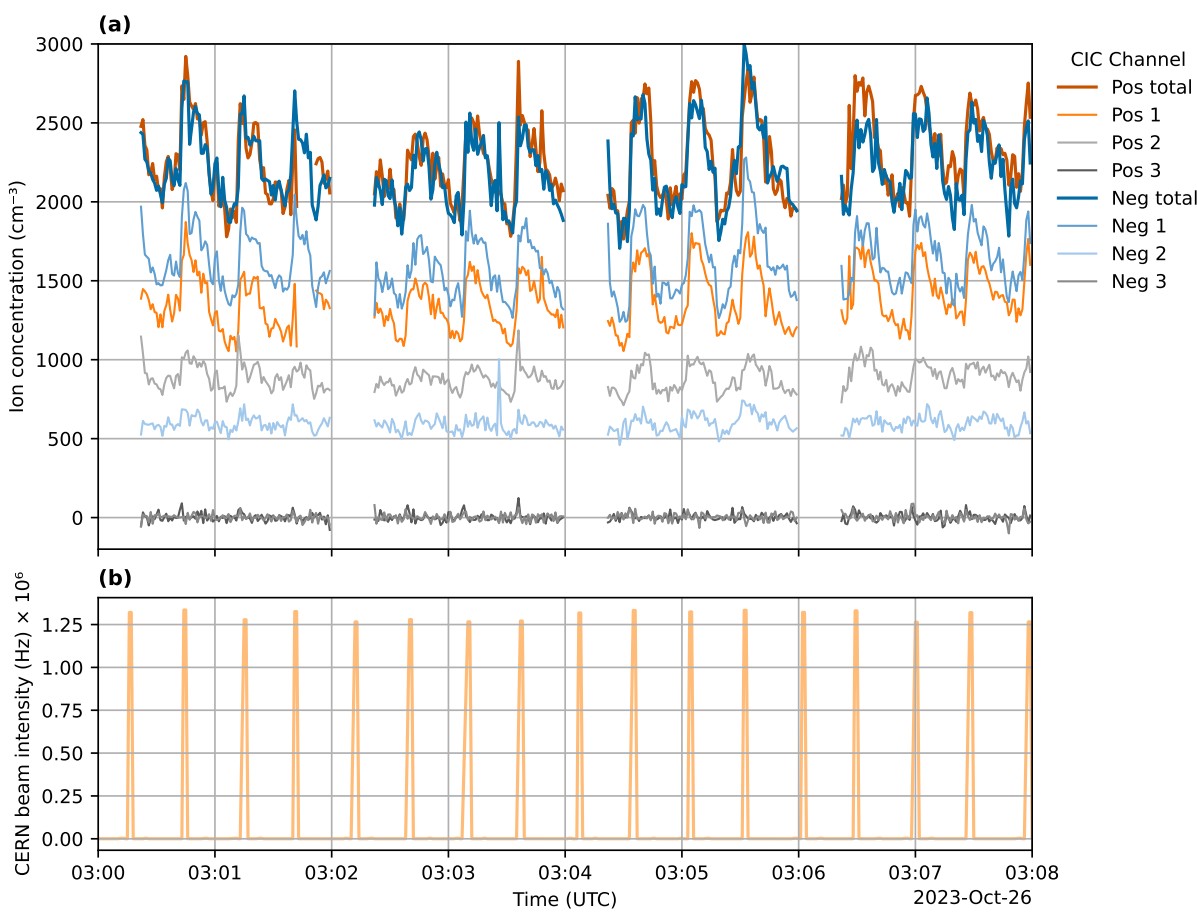

**Figure 10. CIC response to CERN beam pulses.** Evolution over a eight-minute interval of (a) ion number concentrations measured by the CIC when sampling air from the CLOUD chamber at $7\,l\,min^{-1}$ per analyzer and (b) CERN particle beam intensity incident on the chamber. Both panels are displayed versus time at 1 s sampling rate. The CIC reacts with 1 s time response to the pulse of beam ionization and then the ions are diluted as they are mixed with unexposed air in the chamber by internal mixing fans (panel a). The gaps in the CIC data correspond to periods when the device performed automatic zero-level measurements.





**Table 1.** Trace gas, temperature and relative humidity conditions corresponding to the CIC background measurements shown in Fig. 9.

| Measurement | T [°C] | RH [%] | SO2 [ppb] | O3 [ppb] | NO [ppb] | NO2 [ppb] | Other Gases | N [cm$^{-3}$] |
|---|---|---|---|---|---|---|---|---|
| A | -30 | 57 | 0.2 | 3.9 | 0.0 | 0.1 | Isoprene = 0.13 [ppb] | 55.0 |
| B | -49 | 25 | 0.2 | 689.6 | 1.1 | 0.0 | Isoprene = 0.17 [ppb] | 0.9 |
| C | 5 | 45 | 0.3 | 397.9 | 0.0 | 0.0 | Alphapinene = 0.50 [ppb] | 891.9 |
| D | -11 | 76 | 0.2 | 551.3 | 0.0 | 0.0 | Alphapinene = 0.05 [ppb] | 6.4 |
| E | 20 | 48 | 4.8 | 45.7 | 0.0 | 1.2 | SA = $2.1 \times 10^7$ [cm$^{-3}$] | 31.0 |
| F | 20 | 48 | 5.3 | 45.5 | 0.0 | 1.2 | SA = $2.3 \times 10^8$ [cm$^{-3}$] | 6654.2 |
| G | 9 | 15 | 0.3 | 367.3 | 0.1 | 0.0 | DMS = 0.84 [ppb]<br>SA = $1.8 \times 10^6$ [cm$^{-3}$]<br>MSA = $7.6 \times 10^6$ [cm$^{-3}$] | 0.4 |
| H | 9 | 15 | 0.4 | 616.9 | 0.1 | 0.0 | DMS = 2.23 [ppb]<br>SA = $1.5 \times 10^7$ [cm$^{-3}$]<br>MSA = $1.6 \times 10^8$ [cm$^{-3}$] | 130.0 |
| I | -1 | 27 | 0.2 | 500.0 | 0.0 | 0.1 | DMS = 1.83 [ppb]<br>SA = $5.2 \times 10^5$ [cm$^{-3}$]<br>MSA = $6.4 \times 10^6$ [cm$^{-3}$] | 2.1 |
| J | -1 | 27 | 0.3 | 491.1 | 0.0 | 0.0 | DMS = 1.73 [ppb]<br>SA = $8.9 \times 10^5$ [cm$^{-3}$]<br>MSA = $2.1 \times 10^7$ [cm$^{-3}$] | 75.7 |
| K | -11 | 11 | 0.3 | 570.1 | 0.0 | 0.0 | DMS = 1.32 [ppb]<br>SA = $1.5 \times 10^6$ [cm$^{-3}$]<br>MSA = $3.5 \times 10^7$ [cm$^{-3}$] | 6.9 |

Abbreviations: DMS – dimethyl sulfide, SA – sulfuric acid, MSA – methanesulphonic acid.