# Peer review of "Design and performance of the Cluster Ion Counter (CIC)"

_Atmospheric Measurement Techniques, 2024_

## Author Comment (AC1)

**Response to RC1: 'Comment on amt-2024-138', Anonymous Referee #2**

Air ions promote new particle formation through ion-induced nucleation, so measuring air ions, especially small ones, is crucial. This manuscript describes a newly designed instrument, Cluster Ion Counter (CIC), which measures the number concentrations of air ions below 5 nm. Such a device could complement the family of instruments for studying the new particle formation. This manuscript is well written. I believe this manuscript could be published in AMT after addressing the following comments.

We appreciate the reviewer's positive and constructive feedback. Responses to each comment are provided individually below the respective comment, highlighted in blue text.

Major comments:

1. Neutral cluster and Air ion Spectrometer (NAIS), which was designed by same research group of this study, is widely used for the observation of air ions. I'm curious to know what differences or advantages the CIC has over NAIS. This study appears to have done a parallel comparison experiment of CIC and NAIS (Figure 5), but the results are not mentioned in the manuscript.

We understand the reviewer's interest in a direct comparison of NAIS and CIC using CERN CLOUD chamber data. However, such a comparison is not included in our study due to the unique inlet configurations required for sampling from the CLOUD chamber, which would introduce considerable variability and limit the reliability of a direct intercomparison. Specifically, the CLOUD chamber's outlet sizes (1/2 inch or 1 inch) required the use of different adaptors for each instrument (30 mm to 1/2 inch Y-splitter for CIC and 30 mm to 1 inch for NAIS). Additionally, the limited total flow available from the CLOUD chamber, necessitated by the demands of multiple instruments, forced instrument-specific flow rate adjustments. The CIC was operated at a reduced 14 lpm (7 lpm per analyzer), whereas the NAIS, with its fixed sampling rate at 54 lpm, drew 20 lpm from the chamber, with the remaining flow provided by a diluter. These disparities in inlet configurations and flow conditions, leading to potential variations in line losses and introducing uncertainties from either dilution (NAIS) or a potentially lower signal-to-noise ratio (CIC), make a direct and meaningful intercomparison of instrument performance within the CLOUD context difficult. We do, however, draw attention to a recent study by Kulmala (2024) that provides an intercomparison of these instruments under ambient conditions at the SMEAR II station in Finland.

The primary benefits of the CIC over a NAIS are lower weight, smaller dimensions, lower power consumption and better detection efficiency.

We added comments about that to the end of section "2.1 Instrument design":

Compared with earlier instruments developed at the University of Tartu, such as the NAIS or the Balanced Scanning Mobility Analyzer (BSMA; Tammet, 2006), the CIC features a simpler design. While it does not match the sensitivity of the BSMA or provide the detailed sizing information of the NAIS, the CIC offers several advantages over these more complex instruments. Notably, the CIC can operate at low sample flow rates that are unachievable by the BSMA. Compared with the NAIS, the CIC has a higher detection efficiency and a shorter inlet tract, allowing for more reliable measurement of small ions at higher time resolution and with a lower mobility diameter cut-off.

2. CIC looks quite small compared to the NAIS. What is the weight of the CIC? Can mobile observation be an advantage for CIC?

The external dimensions of the CIC are 200 mm width, 130 mm height and 400 mm length. The weight of the instrument is 5 kg. The power consumption is 4-5 W at 30 lpm sample flow rate per analyzer. We added these details to section 2.1.

We added a comment to the summary:

**Its low weight, small dimensions, and low power consumption open up new possibilities for aerial measurements with drones**

3. The authors state that CIC is capable of making precise and robust long-term measurements. Did the CIC operate under ambient conditions before? How does CIC perform in field measurements? I think it is important to make it clear that CIC can be used not only for chamber experiments but also for long-term field measurements.

We appreciate the reviewer's point regarding the broader applicability of the CIC beyond controlled laboratory settings. Indeed, multiple CIC instruments have been deployed for long-term measurements under diverse ambient conditions across various key field locations. For example, CICs have been integral to continuous long-term monitoring at prominent atmospheric research stations, including SMEAR Estonia, SMEAR II in Hyytiälä, Finland, Ny-Ålesund Research Station, and even the challenging and remote environment of Dome C, Antarctica. Kulmala et al. (2024) have already published the first CIC measurements from SMEAR II, covering two and a half months. Forthcoming publications will present measurements from Ny-Ålesund (Vaittinen et al., 2025, conference abstract) and other field sites.

Minor comments:

Abstract: I suggest adding a description of CIC application prospects to the abstract.

We added a short mention about CIC applications to the abstract.

The CIC is primarily designed as a robust, low-maintenance instrument prioritizing ease of operation and broad applicability, including laboratory experiments, long-term unattended field measurements, as well as mobile, airborne and battery-powered setups. The main application of the device is to study temporal development of total cluster ion concentrations while also providing some information about the ion mobility distribution.

Line 146: The full name of NAIS needs to be given here.

The full name was added.

**References:**

Kulmala, M., Tuovinen, S., Mirme, S., Koemets, P., Ahonen, L., Liu, Y., Junninen, H., Petäjä, T., and Kerminen, V.-M.: On the potential of the Cluster Ion Counter (CIC) to observe local new particle formation, condensation sink and growth rate of newly formed particles, Aerosol Research, 2, 291–301, https://doi.org/10.5194/ar-2-291-2024, 2024.

Vaittinen, A., Sarnela, N., Sipilä, M., Brasseur, Z., Boyer, M., Righi, C., Thakur, R., Mazzola, M., and Quéléver, L.: New Particle Formation and Condensable Vapours in an Arctic Site: Ny-Ålesund, EGU General Assembly 2025, Vienna, Austria, 27 Apr–2 May 2025, EGU25-18095, https://doi.org/10.5194/egusphere-egu25-18095, 2025.

---

## Author Comment (AC2)

**Response to RC2: 'Comment on amt-2024-138', Anonymous Referee #3**

This article describes a study of the response of the CIC instrument. The contents has little scientific interest. Nevertheless, the work is carried out competently. Furthermore, since this instrument is used by various groups for basic research, an article such as this one is needed as basic reference. The article is also brief, so publication would be justified once various technical issues are resolved.

**We thank the reviewer for positive and constructive comments. Our responses to these comments are given separately after each comment in blue text.**

The title includes the term design. However, the article contains few design considerations. Why the relatively small flow rates used? Why 3 size ranges? Why the sizes selected? These and many other general issues related to atmospheric studies would presumably have guided the final choice of operational parameters. For instance, the entry filter appears to be a sphere, and the flow past it is likely to separate and become turbulent. No details are given on how this sphere is supported, but its support surely would have an effect on the flow. If the sphere is supported on the inner electrode, then there would be a boundary layer on this support, with a decelerating region near the ogive of the inner electrode, where the boundary layer would separate and increase the level of turbulence. Why would a turbulent flow field be preferable to a laminar one at such low flow rates. How are the flow calculations shown in Figure 3 executed? Either these various key issues are discussed, or the term design should be removed from the title. Even so, the reliability of the mobility response of the device depends on the flow field, so some discussion of this seems inevitable.

The CIC operates with sample flow rates ranging from 5 to 50 l/min per analyzer. The lower limit was selected to accommodate the maximum acceptable sample flow rate for integration with the CERN CLOUD chamber experiment. The upper limit of 50 l/min is determined by the requirement to maintain laminar flow within the instrument under typical operating conditions.

The first two channels of the CIC with limiting mobilities 2.5 and 0.5 cm2/V/s are designed to capture overlapping portions of the complete typical atmospheric cluster ion mobility band. This allows for a rough estimation of the average cluster ion mobility based on the balance of signal between the two channels. The third channel (0.25 cm2/V/s) is set to capture a part of all larger ions to enable the compensation for their contribution to the signals in the first two channels and allowing for a virtual sharp cut-off size. The three channel configuration has additionally facilitated the development of a modified instrument with different limiting mobility values specifically designed for new particle formation studies (Kulmala, 2024).

The inlet sphere is supported by two small, diametrically placed spokes. Although the spokes and the sphere itself inevitably disturb the flow field, both computational fluid dynamics (CFD) models and experimental results indicate that this effect on the instrument's transfer functions is negligible. This is partly because the CIC is designed as a

low-mobility-resolution instrument and the transfer functions are inherently relatively wide and smooth .

CFD calculations were carried out using COMSOL Multiphysics® software. The model agrees that some flow separation and closed circulation occurs behind the sphere but the model does not suggest that the effect reaches further downstream and disturbs the laminar flow pattern in the mobility classification region. The close match between experimentally acquired instrument transfer functions and the idealized mathematical model of a first order differential mobility analyzer (Tammet, 1970) proves that the flow pattern disturbance is not significant for the operation of the device (Figure 5 in revised paper).

Alternative inlet filter designs, such as a mesh where a voltage could be applied, were evaluated, but the spherical inlet filter was selected because it demonstrated the lowest associated particle losses.

There are important design considerations besides the flow field. For instance, why an integrating electrometer rather than a direct current measurement with an inverting amplifier? The one developed by the Burtscher group features a noise level of about 0.1 fA at 1 Hz, and responds in less than 100 ms (i.e. Aerosol Science and Tech., 51(6), 724 – 734, 2017). The fact that zero current level measurement need to be taken every 1-5 minutes suggests that there is a considerable drift, which is not the case in the Burtscher circuit.

The selection of an integrating electrometer for the CIC was primarily based on the long-term proven robustness, reliability and performance of this design in the Neutral cluster and Air Ion Spectrometers (NAIS) which have been in operation in a wide range of demanding environmental conditions for over two decades.

While direct current measurement electrometers, such as the low-noise, fast-response design developed by the Burtscher group, offer similar excellent performance characteristics, the integrating amplifier approach provides the advantage that all electric current arriving between separate analog-digital converter samples is captured. Admittedly, this specific benefit is less critical in the CIC with only 6 channels compared to the NAIS which has 50 channels.

The recommendation for 1-5 minute zero-level measurements considers the instruments typical deployments in both ambient and chamber studies, where rapid temperature fluctuations are common. While the measurement electronics are designed for minimal temperature sensitivity, some thermal drift is unavoidable.

An additional and significant factor influencing the zero level currents, especially during long-term measurements, is the gradual contamination of the collecting electrodes and insulators (e.g., with dust). This contamination can lead to leakage currents, which can be exacerbated by high relative humidity.

Performing frequent zero measurements allows for the compensation of these combined drift and leakage currents. This ensures the reliability of the measurement results, helps to extend the intervals between necessary physical maintenance (cleaning of the analyzer), and provides a clear diagnostic indicator when such maintenance is required. We updated the subsection "2.1 Instrument design" to clarify the points raised by the reviewer regarding instrument design details and rationale:

- The reason for choosing three collecting electrodes and the mobility ranges covered by each.
- The reason for choosing the integrating electrometer.
- The benefits of frequent zero measurements.
- The mechanical design of the inlet filter sphere and its effect on the flow pattern in the analyzer.
- Numerical modelling software used.

We updated section "3.1 Detection efficiency in sub-5 nm diameter range" to emphasize that the close match between experimental results and the simple theoretical model that ignores turbulence and other specifics of the air flows indicates that the flow disturbances in the analyzer are not critical for the device's operation.

The generation of ions with such high mobilities in air (2.5 cm2/V/s) is not a simple matter, and requires some more explanation. We are told they are produced by a hot W wire in N2 and selected by a Herrmann DMA. Would the authors please include a Herrmann DMA mobility spectrum with the ion calibration peaks and some discussion. Is this high mobility cluster a bare metal cation? What other comparably high mobilities are produced by this source? Is the ultradry environment essential for this? Does the neutralizer need to be cleaned specially to yield such high mobilities? Most neutralizers are contaminated and would tend to transform such high mobility ions into larger solvated particles.

We acknowledge that achieving such high mobilities in air requires careful consideration of the ion's composition and the experimental setup. To address this, we provide further information regarding the ions generated by our hot tungsten wire source, which is identical to that used by Kangasluoma et al. (2015). Their work characterized the WOx ions produced by this type of generator using an atmospheric pressure interface time-of-flight mass spectrometer (APi-TOF), both with and without a neutralizer. Table 1 of Kangasluoma et al. (2015) details the negative ions detected. Self-charged clusters were identified as WxOyH ions with an x/y ratio around 1/3 up to at least 2000 amu. In contrast, clusters neutralized by the 241Am sources were WxOy clustered with charge carriers such as NCO, C2H3, NO3 and HNO3NO3. Positive ion identification was more challenging due to organic impurities, with clusters below 600 amu being predominantly organic and heavier clusters consisting of tungsten oxide mixed with organics. Prolonged heating of the wire was found to reduce organic contamination, a finding echoed by Domaschke et al (2019) who used ultra-pure gases and a wire generator without polymer insulators. Peineke (2006) also demonstrated that outgassing at high temperatures significantly impacted positive particle production. Peineke and Schmidt-Ott (2008) further suggest that inherent impurities within the wire material contribute to self-charged positive ion generation through surface ionization, while negative particles are formed by thermo emission of electrons.

Regarding the cleanliness of the neutralizer, Fernandez de la Mora et al. (2003) have shown that even under relatively pure conditions, species can adsorb onto nanoparticle surfaces generated by a radioactive neutralizer, forming a coating whose thickness likely depends on

the charger's history. The 241Am neutralizer used in our setup is dedicated to this experimental line; however, its prior use may have involved neutralizing Ag particles from a furnace or Sodium Chloride / Ammonium sulfate particles from an atomizer. Without in-line APi-TOF measurements downstream of our neutralizer, we cannot definitively ascertain the chemical purity of the neutralized clusters in our experiments. However, as the reviewer points out, significant contamination from the neutralizer would likely shift the observed particle size distribution to larger sizes. This was not observed in our measurements. As shown in Figure R1, clear peaks were recorded at mobilities corresponding to 1.08 nm (1.74 cm²/V/s), 1.37 nm (1.09 cm²/V/s), and a shoulder peak at 1.6 nm (0.8 cm²/V/s). No prominent peaks were detected at larger sizes. Instead, a gradual increase in concentration was observed up to 2.8 nm. While this behaviour above 1.6 nm can be due to contamination, it can also be due to the high temperatures of the wire producing high concentrations of vapors, which consequently produce a continuum of large diameters (Attoui 2022)

We unfortunately do not have a plot for positive polarity measurements, but based on prior work and our observations, the positive mode ions are likely organic species formed due to heat from either the wire or the radioactive source. These impurities tend to dominate in positive polarity due to their higher proton affinities.

We would like to emphasize that the CIC calibrations presented in this manuscript do not require the use of chemically pure metal clusters. Under ambient or chamber conditions, cluster ions are composed of inorganic species (primarily sulfates) and organics, rather than pure metal oxides. Therefore, the chemical identity of the generated particles is of secondary importance for calibration purposes. The glowing wire generator was selected not for the specific composition of the particles it produces, but due to its operational robustness and proven capability to generate nanoparticles across a wide size range and at high number concentrations. Notably, recent work by Attoui (2022) coated the wire surface on purpose with sodium chloride to produce hydrophilic calibration ions in the sub-5 nm range.

Regarding the use of nitrogen: we employed N2 because heating the wire in compressed air caused it to break almost instantly because of the oxygen content.

A dry environment is typically maintained during calibration experiments to ensure stable aerosol generation. The presence of water vapor in the flow can introduce instabilities, as aerosol growth is highly sensitive to relative humidity (RH). Given that particle growth curves are often exponential, even slight variations in RH can result in significant changes in particle size due to hygroscopic growth.

We acknowledge that organic contamination from both the wire generator and the neutralizer can not be ruled out. In positive polarity, it is very likely that the produced ions are some organic impurities generated by the heat of the wire generator or the radioactive source. Small ions produced by the neutralizer may also pass through the DMA. We are not aware of any studies that have characterized WOx generated particles charged in a neutralizer using mass spectrometry that could answer which other high mobilities are produced and whether an ultradry environment is essential. However as the sample is mobility classified using a DMA and CIC only detects ion mobility, the exact cluster composition is not critical for these experiments.

Figure: WOx distribution from CIC calibration, negative polarity

**We updated sections 2.3 and 2.3.1 to clarify the details of the calibration experiments.**

On the same subject of design, I was especially puzzled by the broad remarks on the difficulty to detect the low atmospheric levels of ions, combined with a low flow rate instrument. Given that the Tartu group has previously developed ion detectors with much higher sampling flow rates, what special advantage does the current device offer to compensate for its greatly reduced sensitivity?

We thank the reviewer for raising a valid point regarding the sensitivity trade-off with the CIC compared to previous instruments developed by the Tartu group.

While ion detectors such as the Symmetric Inclined Grid Mobility Analyzer (SIGMA) and the Balanced Scanning Mobility Analyzer (BSMA) do indeed offer higher sensitivity, their very high sample flow rates (exceeding 2000 I/min) make them unsuitable for chamber experiments and most laboratory measurements. Furthermore, their mechanical dimensions, specific inlet requirements and more demanding maintenance routines limit their practicality in field measurements.

The CIC was designed as a robust, low-maintenance instrument prioritizing ease of operation and broad applicability, including for example mobile and battery-powered setups or Eddy covariance measurements.

We updated section 2.1 to include a short comparison to earlier ion instruments developed at the University of Tartu and emphasize the benefit of the low sample flow rate.

**References:**

Kulmala, M., Tuovinen, S., Mirme, S., Koemets, P., Ahonen, L., Liu, Y., Junninen, H., Petäjä, T., and Kerminen, V.-M.: On the potential of the Cluster Ion Counter (CIC) to observe local new particle formation, condensation sink and growth rate of newly formed particles, Aerosol Research, 2, 291–301, https://doi.org/10.5194/ar-2-291-2024, 2024.

Attoui, Michel. 2022. "Mobility Distributions of Sub 5 Nm Singly Self-Charged Water Soluble and Non-Soluble Particles from a Heated NiCr Wire in Clean Dry Air." *Aerosol Science and Technology* 56 (9): 859–68. https://doi.org/10.1080/02786826.2022.2095892.

Domaschke, Maximilian, Christian Lübbert, and Wolfgang Peukert. 2019. "Analysis of Ultrafine Metal Oxide Particles in Aerosols Using Mobility-Resolved Time-of-Flight Mass Spectrometry." *Journal of Aerosol Science* 137 (November):105438. https://doi.org/10.1016/j.jaerosci.2019.105438.

Kangasluoma, J., M. Attoui, H. Junninen, K. Lehtipalo, A. Samodurov, F. Korhonen, N. Sarnela, et al. 2015. "Sizing of Neutral Sub 3nm Tungsten Oxide Clusters Using Airmodus Particle Size Magnifier." *Journal of Aerosol Science* 87 (September):53–62. https://doi.org/10.1016/j.jaerosci.2015.05.007.

Peineke, C., M.B. Attoui, and A. Schmidt-Ott. 2006. "Using a Glowing Wire Generator for Production of Charged, Uniformly Sized Nanoparticles at High Concentrations." *Journal of Aerosol Science* 37 (12): 1651–61. https://doi.org/10.1016/j.jaerosci.2006.06.006.

Peineke, C., and A. Schmidt-Ott. 2008. "Explanation of Charged Nanoparticle Production from Hot Surfaces." *Journal of Aerosol Science* 39 (3): 244–52. https://doi.org/10.1016/j.jaerosci.2007.12.004.

De la Mora, J.F., De Juan, L., Liedtke, K. and Schmidt-Ott, A. 2003. Mass and size determination of nanometer particles by means of mobility analysis and focused impaction. Journal of aerosol science, 34(1), 79–98. https://doi.org/10.1016/S0021-8502(02)00121-0

---

## Author Comment (AC3)

**Response to CC1: 'Atmospheric ion measurements: air conductivity versus ion counting', Karen Aplin, 27 Nov 2024**

The paper esign and performance of the cluster ion counter (CIC) by Mirme et al describes the latest instrument in over fifty years of development of Estonian atmospheric ion spectrometers, begun by the late Prof Hannes Tammet. It seems another excellent instrument, which is well-characterised both theoretically and experimentally. This paper is a carefully and clearly written description that I hope could be further improved with some additions to the introductory material.

The Ebert ion counter and the Gerdien condenser are aspirated condensers, developed at around the same time at the start of the twentieth century (Flagan 1998). In the CIC paper's introduction, the different types of aspirated coaxial cylindrical condenser are listed all together, implying they are essentially identical. There are however some meaningful differences between them. An ion counter, such as that designed by Ebert, operates at a sufficiently high voltage for the electric field in the condenser to collect all the ions passing through the device. In contrast, a Gerdien-type instrument operates in a lower electric field regime, such that only a portion of the ions are collected, which measures atmospheric conductivity rather than counting ions directly (Chalmers 1967). The ion concentration can be estimated from the atmospheric conductivity if a suitable ion mobility can be assumed or separately determined. Understanding the distinctions between these types of instrument is important in interpreting their data.

**We thank Professor Alpin for this valuable clarification. In the revised manuscript, we have removed the text that previously grouped all aspirated coaxial cylindrical condensers together, as it could misleadingly imply that these instruments are essentially identical.**

The paper states that "one limitation of many devices" is their inability to measure bipolar ions, which the CIC avoids by simply having two sampling tubes biased at opposite polarities. The Gerdien condenser can also be operated, as the name suggests, as a capacitor, with a rate of voltage decay that is inversely proportional to the air conductivity. This "voltage decay mode" (Aplin and Harrison 2000) was commonly used in the first half of the twentieth century, and in many radiosonde ascents (Nicoll 2012), because measuring a voltage was simpler than measuring a small current. The voltage decay approach is less frequently used in modern devices but has been exploited in combination with the current measurement approach for self-calibration (Aplin and Harrison 2001). As the operating principle extends to other geometries, this type of instrument is also used in planetary atmospheric electricity, in which context it is known as a "relaxation probe" (Aplin 2013). In the voltage decay mode, a bias voltage is temporarily applied to charge the condenser. It is then released and the capacitor allowed to decay, with a time constant related to the air conductivity. Both positive and negative ions are involved in this process. The form of the decay also provides information on the ion mobility spectrum (Aplin 2005).

**We thank Professor Aplin for highlighting the important historical and modern applications of the voltage decay mode in Gerdien condensers and related instruments. We acknowledge**

that this mode allows for the detection of both positive and negative ions, and we have updated the manuscript to reflect this point more accurately. However, as originally stated, our emphasis was on the limitation that many conventional instruments cannot simultaneously measure positive and negative ions. While the voltage decay mode allows bipolar ion detection over time, it does not provide simultaneous measurements of both polarities. We have revised the relevant section of the text to clarify this distinction.

Finally, it is worth noting that the total ionisation rate near the surface, combining both cosmic rays and natural radioactivity is 10 cm-3s-1, so C.T.R. Wilson was indeed within a factor of two of the modern average.

We corrected the text in the manuscript to reflect that 10 cm-3s-1 is the total ionisation rate near the surface, combining both cosmic rays and natural radioactivity.

**References**

Aplin, K. L. 2005. "Aspirated Capacitor Measurements of Air Conductivity and Ion Mobility Spectra." Review of Scientific Instruments 76(10).

Aplin, K. L., and R. G. Harrison. 2000. "A Computer-Controlled Gerdien Atmospheric Ion Counter." Review of Scientific Instruments 71(8).

Aplin, K. L., and R. G. Harrison. 2001. "A Self-Calibrating Programable Mobility Spectrometer for Atmospheric Ion Measurements." Review of Scientific Instruments 72(8).

Aplin, Karen L. 2013. Electrifying Atmospheres: Charging, Ionisation and Lightning in the Solar System and Beyond. Dordrecht: Springer Netherlands.

Chalmers, John Alan. 1967. Atmospheric Electricity. Second edition. Oxford: Pergamon Press.

Flagan, Richard C. 1998. "History of Electrical Aerosol Measurements." Aerosol Science and Technology 28(4):301–80.

Nicoll, K. A. 2012. "Measurements of Atmospheric Electricity Aloft." Surveys in Geophysics 33(5):991–1057.